# COVID-19 vaccine hesitancy among Ethiopian healthcare workers

Rihanna Mohammed[1], Teklehaimanot Mezgebe Nguse[2], Bruck Messele Habte[3], Atalay Mulu Fentie[3‡]*, Gebremedhin Beedemariam Gebretekle[4,5‡]

1 Africa Medical College, Addis Ababa, Ethiopia, 2 Departments of Radiography, School of Medicine, College of Health Sciences, Addis Ababa University, Addis Ababa, Ethiopia, 3 School of Pharmacy, College of Health Sciences, Addis Ababa University, Addis Ababa, Ethiopia, 4 Institute of Health Policy, Management and Evaluation, University of Toronto, Toronto, Ontario, Canada, 5 Toronto Health Economics and Technology Assessment (THETA) Collaborative, University Health Network, Toronto, Ontario, Canada

‡ These authors are joint senior authors on this work
* atalay.mulu@aau.edu.et

## Abstract

### Introduction

COVID-19 poses significant health and economic threat prompting international firms to rapidly develop vaccines and secure quick regulatory approval. Although COVID-19 vaccination priority is given for high-risk individuals including healthcare workers (HCWs), the success of the immunization efforts hinges on peoples' willingness to embrace these vaccines.

### Objective

This study aimed to assess HCWs intention to be vaccinated against COVID-19 and the reasons underlying vaccine hesitancy.

### Methods

A cross-sectional survey was conducted among HCWs in Addis Ababa, Ethiopia from March to July 2021. Data were collected from eligible participants from 18 health facilities using a pre-tested semi-structured questionnaire. Data were summarized using descriptive statistics and multivariable logistic regression was performed to explore factors associated with COVID-19 vaccine hesitancy. A p<0.05 was considered statistically significant.

### Results

A total of 614 HCWs participated in the study, with a mean age of 30.57±6.87 years. Nearly two-thirds (60.3%) of HCWs were hesitant to use the COVID-19 vaccine. Participants under the age of 30 years were approximately five times more likely to be hesitant to be vaccinated compared to those over the age of 40 years. HCWs other than medical doctors and/or nurses (AOR = 2.1; 95%CI; 1.1, 3.8) were more likely to be hesitant for COVID-19 vaccine. Lack of believe in COVID-19 vaccine benefits (AOR = 2.5; 95%CI; 1.3, 4.6), lack of trust in the government (AOR = 1.9; 95%CI; 1.3, 3.1), lack of trust science to produce safe and

**Competing interests:** The authors have declared that no competing interests exist.

effective vaccines (AOR = 2.6; 95%CI; 1.6, 4.2); and concern about vaccine safety (AOR = 3.2; 95%CI; 1.9, 5.4) were also found to be predictors of COVID-19 vaccine hesitancy.

## Conclusion

COVID-19 vaccine hesitancy showed to be high among HCWs. All concerned bodies including the ministry, regional health authorities, health institutions, and HCWs themselves should work together to increase COVID-19 vaccine uptake and overcome the pandemic.

## Introduction

The COVID-19 pandemic is worldwide public health, social and economic threat for which efforts have been made to prevent and control the spread around the globe. As of November 08, 2021, more than 250 million COVID-19 cases and 5.06 million deaths were reported worldwide and 6.1 million cases and 150,636 deaths in Africa. Similarly in Ethiopia, since the first COVID-19 case was reported on March 13, 2020, about 367,000 cases and 6551 deaths were reported [1, 2]. Similar to the global situation, the pandemic had a significant impact on Ethiopia's economy; in the best-case scenario, the pandemic is predicted to lower GDP by 6.5%; and in the worst-case scenario, the pandemic might reduce GDP by 16.7% [3].

As there are no specific treatment options available for COVID-19 infection other than the World Health Organization (WHO) recommended public health measures, vaccine development has hastened at an unprecedented pace to control the pandemic. Different countries, research institutes, universities and other concerned organizations have been working to urgently develop and deploy safe and effective vaccines as part of the critical intervention of this pandemic [1–3].

The smallpox vaccine against the deadly contagious smallpox virus was the first vaccine developed in 1796 by the British doctor Edward Jenner. Since then, vaccines have saved millions of lives every year [4]. However, the speed for the invention of a vaccine against COVID-19 has been much quicker than any other vaccine invented in the past [5]. Numerous successful vaccines against COVID-19 have already been publicized and were approved for emergency use in some countries within a year. According to WHO's report, there are currently more than 60 COVID-19 vaccine candidates in clinical development and over 170 in the pre-clinical stage [6]. Among other COVID-19 vaccines, Pfizer-BioNTech, Moderna's and Astra Zeneca are the first three vaccines to be approved as safe and effective vaccines for emergency use by WHO's Emergency Use Listing (EUL) and are being distributed and administered across several countries [7]. Ethiopia is one of the 92 countries eligible for donor-funded doses of COVID-19 vaccines through COVAX (COVID-19 Vaccines Global Access), a global initiative aimed at accelerating vaccine development and ensuring equitable access to all countries [6, 8]. As a result, Ethiopia has received the first shipment of 2.2 million doses of AstraZeneca vaccines produced by the Serum Institute of India through COVAX on March 06, 2021. The Ethiopian Ministry of Health prepared guideline for COVID-19 vaccination and launched on March 13, 2021 in a high-level national event held at Eka Kotebe COVID-19 hospital where frontline HCWs were vaccinated to mark the beginning of the vaccination campaign. The main priority of Ethiopia's COVID-19 vaccination program was for frontline HCWs and other vulnerable populations such as the elderly population. Vaccination cascaded to other healthcare facilities given they fulfill the predefined criteria such as establishment of taskforce

to manage the vaccination program, availability of trained nurses to administer the vaccine and pharmacists to monitor any adverse drug events [8].

As COVID-19 vaccines continue to be distributed and administered in many countries including Ethiopia, hesitation towards the vaccine is becoming a challenge and barrier to cover a large proportion of the vulnerable population. In fact, the WHO has identified vaccine hesitancy as one of the top ten threats to global health in 2019 [9, 10]. The SAGE working group has defined vaccine hesitancy as "delay in acceptance or refusal of vaccination despite the availability of vaccination services." Studies showed that vaccine hesitancy is complex and context specific, varying across time, place and vaccine type and influenced by factors such as complacency, convenience and confidence [11]. Hence, the purpose of this study was to assess COVID-19 vaccination hesitancy and associated factors among HCWs working at Addis Ababa healthcare facilities in Ethiopia.

## Methods and materials

### Study area

This study was conducted in Addis Ababa, the capital city of Ethiopia and headquarters of the African Union. The city has an estimated population of more than five million. Administratively, the city is divided into 10 sub-cities and 116 woredas. Addis Ababa hosts 13 public hospitals, 25 private hospitals, 97 public health centers, 179 primary, 458 medium and 343 specialty clinics [12].

### Study design and participants

An institutional-based cross-sectional survey was conducted from March 01 to 10, 2021. Ethiopia got the first shipment of the vaccine on March 6 and vaccination of HCWs was launched on March 13, 2021, which was after our data collection. The source and study population for this study were all health care providers working in public and private healthcare facilities of Addis Ababa. Healthcare providers who were present at the time of data collection and willing to participate were included in the study.

### Sample size and participant recruitment

The sample size was determined using a formula for single population proportion [13], with the assumption that overall 50% HCWs do not intend to receive any the COVID-19 vaccine and a 5% margin of error. Adding a design effect of 1.5, the final sample size was calculated to be 634. We purposively selected a total of 18 hospitals (10 public and eight private hospitals) and predetermined sample size was allocated to all health facilities. Participants who met the study's inclusion criteria were invited to participate, and data collection continued until each health facility achieved the required sample size.

### Data collection instruments and procedure

Data were collected using a self-administered semi-structured questionnaire (S1 File) that was developed by reviewing different literature [9, 11, 14, 15]. The questionnaire has five sections: participants' socio-demographic characteristics, COVID-19 infection history, perceived concern about COVID-19 infection, attitude towards COVID-19 vaccine and perceived concerns towards the vaccine. The questionnaire was pretested on the study sites among 20 HCWs to assess its clarity, applicability, and ability to really assess what the study wanted to measure. Measurements and responses were crosschecked for missed values, irregularities, inconsistencies, and corrective measures were taken as required.

## Data analysis

The collected data were thoroughly reviewed for completeness and consistency before being coded. Epi Info version 7.2.4 was used for data entry and SPSS version 23 was used for data analysis. Descriptive statistics such as mean, frequency and percentage were used to present participants' characteristics. Associations between the outcome variable (COVID-19 vaccine hesitancy) and explanatory variables were tested using multivariable logistic regression. All clinically important variables were included for the multivariable logistic regression model. A $p < 0.05$ was considered statistically significant. The results of all the logistic regression analyses are reported as odds ratios (OR) with 95% confidence intervals (95%CIs).

## Ethical considerations

The study was conducted according to the guidelines of the Declaration of Helsinki and ethical clearance was obtained from Addis Ababa Regional Health Bureau (Protocol #: አ/አ/ጤ/9634/227) and permission was secured from respective healthcare facilities. Written informed consent was obtained from all study participants after explaining the objective of the study. Privacy and confidentiality of collected information were ensured at all levels using de-identification, password-protected computer and storing of questionnaires in a lockable cabinet.

## Results

### Socio-demographic characteristics of the study population

Table 1 shows the socio-demographic characteristics of the participants. A total of 614 HCWs (with a response rate of 96.8%) participated in this study. Most (350, 57.0%) of the participants were <30 years old (mean age of 30.57 ± 6.87 years), unmarried (357, 58.1%) and Orthodox Christians (383, 62.6%). The majority (488, 79.5%) of participants were first-degree holders with 1–5 years of work experience (394, 64.2%) and were working in the Emergency (143, 23.3%) and medical wards (124, 20.2%). Only a few (60, 9.8%) of them reported that they had chronic medical conditions, of which diabetes accounted for the greater proportion (23, 39.0%) followed by hypertension (10, 16.9%).

### Exposure to COVID-19 infection

As indicated in Table 2, the majority (434, 70.7%) of the HCWs had contact with COVID-19 infected patients while 348 (56.8%) of them stated that they had contact with COVID-19 infected family members/friends. More than half (332, 54.2%) of the participants had prior experience of caring for or treating COVID-19 patients. Despite their exposure, most HCWs (435, 71.2%) who took part in this study did not contract COVID-19.

### Healthcare providers perceived worries about COVID-19 pandemic

Table 3 summarizes the HCWs perceived worries about the COVID-19 pandemic. Due to the nature of their work at the hospital during the COVID-19 pandemic, 549 (91.9%) of respondents were either somewhat or extremely worried about their health. The majority (333, 55.1%) of HCWs reported that they were extremely worried about the potential risk of COVID-19 to their family, loved ones or others as a result of their roles in the hospital than they were about their own health (174, 29.1%). Regarding the potential risk of becoming infected with COVID-19 due to their roles in the hospital, most were somewhat worried (350, 57.9%) followed by extremely worried (229, 37.9%).

**Table 1. Socio-demographic characteristics of study participants.**

| Variables | N (%) |
|---|---|
| **Age (in years)** | |
| < 30 | 350 (57.0) |
| 30–40 | 211 (34.4) |
| >40 | 53 (8.6) |
| **Gender** | |
| Male | 298 (48.5) |
| Female | 316 (51.5) |
| **Marital status** | |
| Married | 257 (41.9) |
| Unmarried | 357 (58.1) |
| **Religion** | |
| Orthodox Christian | 383 (62.6) |
| Muslim | 102 (16.7) |
| Protestant Christian | 102 (16.7) |
| Catholic Christian | 8 (1.3) |
| Others[a] | 17 (2.7) |
| **Working institution** | |
| Government hospital | 429(69.9) |
| Private Hospital | 185(30.1) |
| **Profession type** | |
| Medical doctors | 168 (27.4) |
| Nurses and Midwives | 257 (41.9) |
| Other HCWs[b] | 189 (30.8) |
| **Educational status in health sciences** | |
| Diploma | 39 (6.4) |
| First degree | 488 (79.5) |
| Postgraduate | 87 (14.2) |
| **Work experience (in years)** | |
| <5 | 394 (64.2) |
| 6–10 | 154 (25.1) |
| >10 | 66 (10.7) |
| **Primary work unit** | |
| Emergency | 52 (8.5) |
| COVID-19 management unit | 91(14.8) |
| Surgical ward | 31 (5.0) |
| Medical ward | 124 (20.2) |
| Intensive care unit | 37 (6.0) |
| Gynecology and Obstetrics ward | 81 (13.2) |
| Others[c] | 198 (32.2) |
| **Presence of confirmed chronic illness** | |
| Yes | 60 (9.8) |
| No | 554 (90.2) |
| **Type of chronic illness** | |
| Hypertension | 10 (16.9) |
| Diabetes Mellitus | 23 (39.0) |

(*Continued*)

**Table 1.** (Continued)

| Variables | N (%) |
|---|---|
| Others[d] | 27 (44.1) |

[a]Atheist, Waqefetta,

[b]Anesthetic technician, Medical laboratory technologists, Pharmacists, X-ray technicians,

[c]Imaging unit, Laboratory, Pharmacy,

[d]Congestive heart failure, Chronic Asthma, Psychiatric illness

## Attitude of healthcare workers towards COVID-19 vaccine

As shown in Table 4, a total of 416 (67.9%) participants reported that they had received another vaccine at their adult age. When asked if they would be willing to be vaccinated when the vaccine became accessible, 65% said they would be certainly or probably willing. In terms of timing, approximately 40% of respondents indicated that they would be willing to be vaccinated as soon as it became available. However, roughly 38% of HCWs preferred to wait a few months before taking the vaccine. Likewise, nearly 38% of participants intended to recommend the vaccine to their patients/ clients and the general public, whereas about 12% of them were unwilling to do so because they believed that COVID-19 shots vaccines are harmful and worthless.

Most of the participants (275, 45.2%) agreed that acquiring natural immunity against infectious disease (by contracting the disease) is better than vaccination. Almost a quarter (163, 26.7%) of participants wrongly believed that COVID-19 vaccines contain live viruses that may cause infection. During the survey, more than half (342, 56.0%) HCWs did not get training on COVID-19. On the other hand, HCWs' decision to take or not take the vaccination was mostly influenced by information obtained from electronic or printed media (244, 39.8%) followed by vaccine producers (164, 26.8%).

## Perceived concerns of COVID-19 vaccine

A summary of HCWs' perceived concerns about the COVID-19 vaccine are presented in Table 5. Only a few HCWs agreed that the currently available COVID-19 vaccines are safe (124, 20.2%) and efficacious (117, 19.2%). About half (306, 50.0%) of respondents were concerned about the vaccines' safety and 279(45.7%) respondents voiced their concern about the efficacy of the vaccines. Nearly half of HCWs were concerned about the potential short- and

**Table 2. Exposure status of healthcare providers to COVID-19 infection.**

| Variable | Response | N (%) |
|---|---|---|
| Previous contact with COVID-19 infected patient | Yes | 434 (70.7) |
| | No | 101 (16.4) |
| | Not sure | 79 (12.9) |
| Previous contact with COVID-19 positive family member or friend | Yes | 348 (56.8) |
| | No | 233 (38.0) |
| | Not remembered | 32 (5.2) |
| Caring and/or treating COVID-19 patients | Yes | 332 (54.2) |
| | No | 280 (45.8) |
| Being infected with laboratory confirmed COVID-19 disease | Yes | 176 (28.8) |
| | No | 435 (71.2) |

**Table 3. Healthcare providers perceived worries about COVID-19 pandemic.**

| Variable | Response | N (%) |
|---|---|---|
| Extent of worry about personal health due to roles in the hospital during COVID-19 pandemic | Extremely worried | 174 (29.1) |
| | Somewhat worried | 375 (62.8) |
| | Not worried at all | 48 (8.0) |
| Extent of worry about the potential risk of becoming infected with COVID-19 due to roles in the hospital | Extremely worried | 229 (37.9) |
| | Somewhat worried | 350 (57.9) |
| | Not worried at all | 25 (4.1) |
| Extent of worry about the potential risk of COVID-19 to one's family, loved ones or others due to roles in the hospital | Extremely worried | 333 (55.1) |
| | Somewhat worried | 242 (40.1) |
| | Not worried at all | 29 (4.8) |

long-term negative effects of the vaccine, while 26% were concerned about the risk of COVID-19 infection as a result of the vaccine. Most (254, 41.5%) of the respondents agreed that the COVID-19 vaccines' adverse effects are acceptable to them. Yet, 216(35.3%) were unsure whether the adverse effects of the vaccines are acceptable or not. The majority (317, 51.9%) of HCWs stated that they have no medical contraindications to COVID-19 vaccines. More than half (58.4%) of respondents said they trust science to develop safe and effective vaccines. Aside from that, 244 (39.9%) of respondents were unsure whether they trusted the Ministry of Health to assure vaccine safety.

## Factors associated with COVID-19 vaccine hesitancy

As shown in Table 6, the logistic regression analysis revealed that age, years of work experience, profession type, working department/unit, belief in the benefits of COVID-19 vaccine, perception of naturally acquired immunity versus vaccine immunity, trust in the Ministry of Health, trust in science, perception of COVID-19 vaccine safety, and perceived risks of COVID-19 vaccine were significantly associated with COVID-19 vaccine hesitancy. However, sex, marital and educational status, history of confirmed COVID-19 infection and presence of comorbidities were not associated with vaccine hesitancy.

When compared to those aged >40 years, HCWs aged <30 years were more than five times (AOR = 5.3; 95%CI; 2.0, 14.1) more likely to be hesitant to COVID-19 vaccine. Similarly, HCWs between the age of 30–40 years were about four times (AOR = 4.7; 95%CI; 1.9, 11.9) more likely to be hesitant towards COVID-19 vaccine compared to their counterparts. HCWs other than medical doctors and/or nurses (AOR = 2.1; 95%CI; 1.1, 3.8) were more hesitant to COVIDD-19 vaccine compared to medical doctors. Those with ≥10 years of work experience (AOR = 0.40; 95%CI; 0.1, 0.9), and those who had been working at the Emergency Department (AOR = 0.3; 95%CI; 0.1, 0.9) were less hesitant to receive the COVID-19 vaccine compared to their counterparts. Healthcare providers who agreed with the statement "I do not believe COVID-19 vaccine will benefit me because I have strong immunity" were more likely to be hesitant to COVID-19 vaccine compared to their peers (AOR = 2.5; 95%CI; 1.3, 4.6). Likewise, hesitancy to COVID-19 vaccine was higher among HCWs who agreed with the statement "acquiring immunity naturally (by contracting the disease) is better than via vaccination) (AOR = 1.6; 95%CI; 1.1, 2.4).

Participants who did not trust the Ministry of Health to assure the safety of COVID-19 vaccine were nearly two times (AOR = 1.9; 95%CI; 1.3, 3.1) more likely to be hesitant of taking the vaccine than their counterparts. Higher odds of vaccine hesitancy (AOR = 2.6; 95%CI; 1.6, 4.2) was also found among participants who do not trust science to produce safe and effective

**Table 4. Attitude of healthcare providers towards COVID-19 vaccine.**

| Variable | Response | N (%) |
|---|---|---|
| Willingness to be vaccinated with the COVID-19 vaccine when it becomes accessible | Yes, certainly | 227 (37.0) |
| | Yes, probably | 172 (28.0) |
| | No, probably not | 107 (17.4) |
| | No, certainly not | 54 (8.8) |
| | Do not know | 54 (8.8) |
| Time to take the COVID-19 vaccine | I will take a shot as soon as possible | 242 (39.4) |
| | I will delay getting it for a few months | 236 (38.4) |
| | I will never take the vaccine | 136 (22.2) |
| Willingness to recommend the COVID-19 vaccine once it is accessible to the public to patients/clients and other community members | Yes, certainly | 230 (37.5) |
| | Yes, probably | 219 (35.7) |
| | No, probably not | 71 (11.6) |
| | No, certainly not | 41 (6.7) |
| | Do not know | 53 (8.6) |
| Belief about COVID-19 vaccines | COVID-19 vaccine is the most likely way to stop this pandemic | 241 (42.4) |
| | The best way to avoid the complications of COVID-19 is to be vaccinated | 260 (45.8) |
| | COVID-19 vaccines are harmful and useless | 67 (11.8) |
| It is preferable to acquire immunity against infectious disease naturally (by having the disease) than by vaccination | Agree | 275 (45.2) |
| | Disagree | 233 (38.3) |
| | Do not know | 100 (16.4) |
| Best sources of information about COVID-19 and its vaccines? | Social media | 326 (53.4) |
| | The government websites | 116 (19.0) |
| | Television/Radio | 54 (8.9) |
| | Telecommunication (text and voice message) | 13 (2.1) |
| | Peer | 30 (4.9) |
| | Religious place | 16 (2.6) |
| | Others[a] | 55 (9.0) |
| Ever receipt of any vaccine at adult age | Yes | 416 (67.9) |
| | No | 156 (25.4) |
| | Not sure | 41 (6.7) |
| The COVID-19 vaccine contains live viruses that may cause some people to get COVID-19 disease | Agree | 163 (26.7) |
| | Disagree | 230 (37.6) |
| | Unsure | 218 (35.7) |
| Getting vaccinated against COVID-19 is important to protect patients | Agree | 380 (62.6) |
| | Disagree | 89 (14.7) |
| | Unsure | 138 (22.7) |
| Most influencing factor to take or not to take the COVID-19 vaccine | Religious views | 76 (12.4) |
| | Information or advice from peers | 95 (15.5) |
| | Information from vaccine producers | 164 (26.8) |
| | Information from media[b] | 244 (39.8) |
| | Others[c] | 34 (5.5) |
| Received training on COVID-19 | Yes | 269 (44.0) |
| | No | 342 (56.0) |

[a]Information obtained from Peer-reviewed research papers,

[b]Electronic such as TV, Radio, social media and printed or electronic newspapers,

[c]Already infected and improved, perceived vulnerability in contracting the infection

**Table 5. Perceived concerns of healthcare providers regarding COVID-19 vaccine.**

| Variable | Response | N (%) |
| --- | --- | --- |
| Currently available COVID-19 vaccines are safe | Agree | 124 (20.2) |
| | Disagree | 106 (17.3) |
| | Unsure | 384 (62.5) |
| Currently available COVID-19 vaccines are effective | Agree | 117 (19.2) |
| | Disagree | 84 (13.8) |
| | Unsure | 407 (66.9) |
| Concerned about the safety of the COVID-19 vaccine | Yes | 306 (50.0) |
| | No | 79 (12.9) |
| | I need more information on the safety of COVID-19 vaccine before a decision is made as to whether to receive it or not. | 227 (37.1) |
| Concerned about the efficacy of the COVID-19 vaccine | Yes | 279 (45.7) |
| | No | 95 (15.5) |
| | Need more information before a decision is made as to whether to receive it or not. | 238 (38.8) |
| If you are concerned about the safety and efficacy of the vaccine, what kind of risks are you concerned about? | It may not provide short- and/or long-term protection | 97 (25.5) |
| | The potential short- and long-term side effects | 185 (48.6) |
| | Risk of COVID-19 infection due to COVID-19 vaccine itself | 99 (26.0) |
| The side effects of COVID-19 vaccine are not acceptable to me | Agree | 142 (23.2) |
| | Disagree | 254 (41.5) |
| | Unsure | 216 (35.3) |
| I don't believe that COVID-19 immunization will benefit me because I have good immunity | Agree | 95 (15.5) |
| | Disagree | 356 (58.0) |
| | Unsure | 163 (26.5) |
| Presence of medical contradiction to COVID-19 vaccine | Yes | 96 (15.7) |
| | No | 317 (51.9) |
| | Unsure | 198 (32.4) |
| I trust science to develop safe and effective vaccines | Yes | 358 (58.4) |
| | No | 70 (11.4) |
| | Not sure | 186 (30.2) |
| I trust the ministry of health to ensure the safety of COVID-19 vaccine | Yes | 225 (36.8) |
| | No | 143 (23.4) |
| | Not sure | 244 (39.9) |

vaccines. Healthcare providers who disagreed with the statement that "COVID-19 vaccines are safe" were more likely to be hesitant to COVID-19 vaccine (AOR = 3.2; 95%CI; 1.9, 5.4) than their counterparts. Higher odds (AOR = 1.5; 95%CI; 1.1, 2.3) of hesitancy was also found for HCWs who expressed their concern about the risks of COVID-19 vaccine.

## Discussion

The COVID-19 pandemic became a serious threat to the world ever since WHO declared it as a global pandemic on the 11[th] of March 2020. Since then, efforts have been made to control the pandemic from spreading and causing severe illness and death toll. Vaccine development was part of this endeavor, and the COVID-19 vaccine was distributed to Ethiopia and other LMICs under the COVAX worldwide program [16]. However, vaccine hesitancy remains a major public health problem and becoming a barrier to the prevention and containment of the pandemic [9, 14, 17]. This study aimed to assess healthcare professionals' COVID-19 vaccine hesitancy and identify factors aggravating vaccine hesitancy in Ethiopia.

**Table 6.  Variables associated with COVID-19 vaccine hesitancy.**

| Variable | Hesitant to take vaccine, N (%) | | COR (95%CI) | AOR (95%CI) |
|---|---|---|---|---|
| | No | Yes | | |
| **Sex** | | | | |
| Male | 108(44.6) | 208(55.9) | 1.00 | 1.00 |
| Female | 134(55.4) | 164(44.1) | 1.6(1.1,2.2)* | 1.3(0.8, 1.9) |
| **Marital status** | | | | |
| Married | 109(45.0) | 148(39.8) | 1.00 | 1.00 |
| Unmarried | 133(55.0) | 224(60.2) | 1.2(0.9, 1.7) | 1.1(0.7, 1.7) |
| **Age category** | | | | |
| < 30 years | 125(51.7) | 225(60.5) | 3.0(1.6, 5.4) | 5.3(2.0, 14.1)* |
| 30–40 years | 84(34.7) | 127(34.1) | 2.5(1.3, 4.6) | 4.7(1.9, 11.9)* |
| >40 years | 33(13.6) | 20(5.4) | 1.00 | 1.00 |
| **Level of education** | | | | |
| Diploma | 13(5.0) | 26(7.0) | 1.00 | 1.00 |
| First degree | 190(79.5) | 298(80.6) | 0.8(0.4, 1.5) | 0.8(0.3, 1.8) |
| Postgraduate | 39(15.5) | 48(12.4) | 0.7(0.3, 1.3) | 1.3(0.5, 3.6) |
| **Years of work experience** | | | | |
| 1–5 | 147(61.3) | 247(67.9) | 1.00 | 1.00 |
| 6–10 | 68(28.3) | 86(23.6) | 0.75(0.52,1.10) | 0.58(0.24, 1.45) |
| >10 | 27(10.4) | 39(8.5) | 0.74(0.42,1.30) | 0.40(0.20,0.98)* |
| **Profession** | | | | |
| Medical doctors | 85(22.8) | 83(22.3) | 1.00 | 1.00 |
| Nurse and midwives | 91(24.5) | 166(44.6) | 1.9(1.3, 2.8) | 1.2(0.7, 2.0) |
| Other HCWs [a] | 66(17.7) | 123(33.1) | 1.9(1.2, 2.9) | 2.1(1.1, 3.8)* |
| **Working area** | | | | |
| Surgical ward | 9(3.7) | 22(5.9) | 1.00 | 1.00 |
| Medical ward | 51(21.1) | 73(19.6) | 0.6(0.3, 1.4) | 0.6(0.2, 1.5) |
| Intensive care unit | 17(7.0) | 20(5.4) | 0.5(0.2, 1.3) | 0.4(0.1, 1.5) |
| COVID-19 management unit | 30(12.4) | 61(16.4) | 0.8(0.3, 2.0) | 0.7(0.3, 2.1) |
| Gynecology/Obstetrics | 33(13.6) | 48 (12.9) | 0.6(0.2, 1.5) | 0.4(0.1, 1.1) |
| Emergency | 23(9.5) | 29(7.8) | 0.5(0.2, 1.3) | 0.3(0.1, 0.9)* |
| Others[b] | 79(32.6) | 119(31.9) | 0.6(0.3, 1.4) | 2.1(1.1, 3.8)* |
| **History of laboratory confirmed COVID-19 infection** | | | | |
| Yes | 75(31.1) | 101(27.3) | 1.00 | 1.00 |
| No | 166(68.9) | 269(72.7) | 1.2(0.8,1.7) | 1.3(0.9,2.1) |
| **Do not believe COVID-19 vaccine will benefit me** | | | | |
| Agree | 23(9.5) | 72(19.5) | 2.3(1.4,3.8) | 2.5(1.3, 4.6)* |
| Disagree | 219(90.5) | 298(80.5) | 1.00 | 1.00 |
| **Acquiring immunity naturally (by contracting the disease) is better than by vaccination** | | | | |
| Agree | 90(37.5) | 185(50.3) | 1.7(1.2,2.4) | 1.6(1.1, 2.4)* |
| Disagree | 150(62.5) | 183(49.7) | 1.00 | 1.00 |
| **Trust in Ministry of health on the safety of the vaccine** | | | | |
| Yes | 133(55.2) | 93(24.8) | 1.00 | 1.00 |
| No | 108(44.8) | 279(75.2) | 3.7(2.6,5.3) | 1.9(1.3, 3.1)* |
| **Trust in science to develop safe and effective vaccine** | | | | |
| Yes | 185(76.5) | 174(46.6) | 1.00 | 1.00 |
| No | 57(23.5) | 198(53.4) | 3.7(2.6,5.3) | 2.6(1.6, 4.2)** |
| **COVID vaccines are safe** | | | | |

*(Continued)*

**Table 6.** (Continued)

| Variable | Hesitant to take vaccine, N (%) | | COR (95%CI) | AOR (95%CI) |
|---|---|---|---|---|
| | No | Yes | | |
| Agree | 85(35.1) | 39(10.5) | 1.00 | 1.00 |
| Disagree | 157(64.9) | 333(89.5) | 4.6(3.0,7.1) | 3.2(1.9, 5.4)** |
| Concern about the risk of COVID vaccine | | | | |
| Yes | 148(61.2) | 255(68.4) | 1.4(0.9,1.9) | 1.5(1.1, 2.3)* |
| No | 94(38.8) | 117(31.6) | 1.00 | 1.00 |

[a]Anesthetic technician, Medical laboratory technologists, Pharmacists, X-ray technicians,

[b]Pharmacy, laboratory, imaging unit;

*$p<0.05$,

**$p<0.0001$

Healthcare providers play a critical role in the control and prevention of COVID-19 as frontline workers in treating patients, providing information on preventive measures as well as setting themselves as role models to the community. Despite the fact that HCWs were given priority for vaccination due to their increased risk of infection, a significant COVID-19 vaccine hesitancy was reported in many countries which ranged from 8% in the USA to 72.7% in Democratic Republic of Congo [18–27]. Similarly, in our survey, COVID-19 vaccine hesitancy among HCWs was 60.9% which is almost similar to studies conducted among HCWs in the Southern Ethiopia [24] and the general Ethiopian population (68.6%) [25]. Furthermore, when compared to studies conducted in the USA, Saudi Arabia, and China [18, 20, 23], the higher prevalence of COVID-19 vaccine hesitancy observed in our study could be explained by the fact that the vast majority HCWs were unsure about the vaccine's safety and effectiveness and they preferred natural immunity over COVID-19 vaccines.

The most influencing factors to take or not to take the COVID-19 vaccine in our study were different pieces of information from different media platforms (39.8%), vaccine producers (26.8%), information or advice from pears (15.5%) and religious views (12.4%). The lack of effective information communication strategies could be damaging as the vast majority of study participants stated that social media was their primary source of information, which could expose them to false and misleading information. This is supported by the fact that almost a quarter of participants wrongly believed that COVID-19 vaccines contain live viruses capable of causing infection [28–31].

Compared to a study done in the USA (69%) [20], relatively higher concern on the safety of COVID-19 vaccine (89.5%) was reported in this study. However, our finding mirrors with studies conducted in the USA, China, Democratic Republic of Congo, and Malta that long and short-term side effects and efficacy of the vaccine, and the possibility to get COVID-19 infection from the vaccine itself associated with the speed at which the vaccines produced were mentioned by the participants as main concerns and additional reasons for vaccine hesitancy. On the other hand, factors such as lack of awareness about the vaccines, inadequate training and communication about the COVID-19 vaccines safety and efficacy by the concerned bodies were reported as factors contributing to vaccine hesitancy [18–27]. Hence, building trust regarding the ability of the government and other concerned bodies is crucial. Thus, to increase vaccine uptake and acceptance, the Ethiopian Ministry of Health and other concerned bodies should build strategies such as organize intercultural health advocating sessions for HCWs and the community, increasing knowledge and skill of health advocators in terms of vaccine information and interpersonal communication, engage key community leaders in the

information provision, engage vaccine users (HCWs and the community) in providing agreed vaccination information and make informed decisions, engage HCWs in an empathic way and design different vaccine related information communication platforms [32, 33].

Our study showed that younger age and less experienced HCWs had higher odds of vaccine hesitancy compared to those relatively older and more experienced professionals. This could be owing to the active engagement of young HCWs in various social media platforms, which are mostly disseminating negative rumors from unreliable sources [26] and the perceived lesser vulnerability to the infection [20, 30, 31]. In our study, the vast majority of study participants stated that social media was their primary source of information, which could expose them to false and misleading information. Indeed, misleading information regarding COVID-19 vaccinations is quickly circulating on social media, and these erroneous comments about COVID-19 vaccines can have a significant impact on ongoing vaccination campaigns and can pose a threat to global public health [28, 30, 31].

The present study revealed that participants' profession and working area/unit were associated with COVID-19 vaccine hesitancy. HCWs other than medical doctors and/or nurses (i.e. laboratory technologists, radiography technologists, and pharmacy professionals) were found to be more hesitant to take COVID-19 vaccine. In contrary, in another study, nurses had a higher vaccine hesitancy or lower intention to be vaccinated than physicians [25]. Likewise, those who have been working in the pharmacy, laboratory and imaging were more hesitant to get the vaccine. On the other hand, HCWs in the emergency department were more willing to be vaccinated. Our findings mirror other studies that those who had a relatively lower direct patient contact have a reduced perceived risk of COVID-19 infection exposure and worrisome [20, 22, 25, 27].

Our study has some limitations. Due to the cross-sectional nature of the study, a causal relationship between COVID-19 vaccine hesitancy and predictors cannot be established. The study participants were also selected using purposive sampling and there might be a selection bias. However, we recruited a large sample size that will improve the power of accuracy. Moreover, our study included HCWs working at various units of both public and private healthcare facilities that is something that has not been thoroughly researched in Ethiopia. Despite these limitations, this study addresses a timely and relevant researched question that has received little attention in Ethiopia and other LMICs. Thus, the findings can be used as an input to devise interventional strategies to minimize vaccine hesitancy and boost uptake by HCWs who are the gatekeepers in the healthcare system, which ultimately could have a significant impact on COVID-19 prevention and containment in the country.

## Conclusions

The overall COVID-19 vaccine hesitancy among HCWs in Addis Ababa, Ethiopia was found to be high compared to studies reported elsewhere but comparable to vaccine hesitancy among the general population in Ethiopia. Being younger age, HCWs other than medical doctors and/or nurses, belief regarding acquired immunity is superior than vaccination, negative perception on the safety of COVID-19 vaccine, lack of trust in science to produce safe and effective vaccines, lack of trust in the Ministry of Health and concern about risks of COVID-19 vaccine were identified as significant factors contributing to increased hesitancy against COVID-19 vaccine. The intention to take COVID-19 vaccine was higher among HCWs working in the Emergency department but lower in those working in the pharmacy, laboratory and imaging units.

## Supporting information

**S1 File. Data collection tool.**
(DOCX)

**S2 File. A minimally anonymized data set.**
(XLSX)

## Acknowledgments

The authors would like to thank all the study participants for their time and willingness to participate in the study. We also would like to express our sincere gratitude to the respective healthcare facility managers and the data collectors for their assistance throughout the study period.

## Author Contributions

**Conceptualization:** Rihanna Mohammed, Atalay Mulu Fentie, Gebremedhin Beedemariam Gebretekle.

**Data curation:** Rihanna Mohammed, Atalay Mulu Fentie.

**Formal analysis:** Atalay Mulu Fentie.

**Investigation:** Rihanna Mohammed, Atalay Mulu Fentie, Gebremedhin Beedemariam Gebretekle.

**Methodology:** Rihanna Mohammed, Teklehaimanot Mezgebe Nguse, Bruck Messele Habte, Atalay Mulu Fentie, Gebremedhin Beedemariam Gebretekle.

**Supervision:** Teklehaimanot Mezgebe Nguse, Bruck Messele Habte, Atalay Mulu Fentie, Gebremedhin Beedemariam Gebretekle.

**Validation:** Bruck Messele Habte, Atalay Mulu Fentie, Gebremedhin Beedemariam Gebretekle.

**Visualization:** Rihanna Mohammed, Teklehaimanot Mezgebe Nguse, Atalay Mulu Fentie, Gebremedhin Beedemariam Gebretekle.

**Writing – original draft:** Rihanna Mohammed, Atalay Mulu Fentie.

**Writing – review & editing:** Rihanna Mohammed, Teklehaimanot Mezgebe Nguse, Bruck Messele Habte, Atalay Mulu Fentie, Gebremedhin Beedemariam Gebretekle.

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
