## [Decision Letter · Decision Letter 0]

5 Nov 2021

PONE-D-21-31552COVID-19 Vaccine Hesitancy among Ethiopian Healthcare WorkersPLOS ONE

Dear Dr. Fentie,

Thank you for submitting your manuscript to PLOS ONE. After careful consideration, we feel that it has merit but does not fully meet PLOS ONE’s publication criteria as it currently stands. Therefore, we invite you to submit a revised version of the manuscript that addresses the points raised during the review process.

We look forward to receiving your revised manuscript.

Kind regards,

Stephan Doering, M.D.

Academic Editor

PLOS ONE

2. PLOS ONE does not copy edit accepted manuscripts (https://journals.plos.org/plosone/s/criteria-for-publication#loc-5). To that effect, please ensure that your submission is free of typos and grammatical errors.

3. We note that you have indicated in your manuscript that data from this study are available upon request. PLOS only allows data to be available upon request if there are legal or ethical restrictions on sharing data publicly. For information on unacceptable data access restrictions, please see http://journals.plos.org/plosone/s/data-availability#loc-unacceptable-data-access-restrictions. 

Reviewers' comments:

Reviewer's Responses to Questions

**Comments to the Author**

1. Is the manuscript technically sound, and do the data support the conclusions?

Reviewer #1: Yes

Reviewer #2: Yes

2. Has the statistical analysis been performed appropriately and rigorously? 

Reviewer #1: Yes

Reviewer #2: Yes

3. Have the authors made all data underlying the findings in their manuscript fully available?

Reviewer #1: Yes

Reviewer #2: No

4. Is the manuscript presented in an intelligible fashion and written in standard English?

Reviewer #1: Yes

Reviewer #2: Yes

5. Review Comments to the Author

Reviewer #1: General Comments: The manuscript is well-structured overall. Minor grammatical errors can be seen throughout. It is suggested to run the manuscript over some grammar checking app or website like Grammarly.

Specific Comments:

Abstract: In the line 36, change the word 'nurse' to 'nurses'

Methods: In the line 109, change the word 'facilitates' to 'facility'

Results: In the line 144, change 'medical condition' to 'medical conditions'

In the line 182, change 'printed medias' to 'printed media'

In the line 189, change the phrase 'Only few' to 'Only a few'

In the line 229, change the phrase 'trust in science' to 'trust science'

Discussion: Spelling mistakes and grammatical errors need correction

In the line 261, change 'different informations' to 'different pieces of information'

In the line 268, rephrase the sentence 'A relatively higher concerns'

In the line 272, correct the spelling of 'speed'

In the line 275, change 'inadequately communication' to 'inadequate communication'

In the line 277 and line 300, correct the spelling of 'hesitancy'

In the line 284, correct the spelling of 'empathetic'

In the line 299, change 'Incontrary' to 'In contrary'

In the line 303, change the word 'mirrors' to 'mirror'

In the line 311, change the word 'limitation' to 'limitations'

Conclusion: Well-written overall but few grammatical mistakes need correction

In the line 321, change the word 'belief' to 'believe' and change the phrase 'superior over vaccination' to 'superior to vaccination'

Reviewer #2: The author states: “Healthcare providers who were present at the time of data collection and willing to participate were included in the study.” Was data collected once or more than one in the same location? On particular days/times, etc? Further information is needed.

Indicate what table the results are in before describing them (e.g., Table 6 results).

Were vaccines not yet available to Ethiopia from March to July 2021? It would be useful to describe a little bit more about the context, since the questions were asking about whether one planned to receive a vaccine versus whether they had received a vaccine.

6. PLOS authors have the option to publish the peer review history of their article (what does this mean?). If published, this will include your full peer review and any attached files.

Reviewer #1: **Yes: **Qasim Mehmood

Reviewer #2: No

---

## [Author Response · Author response to Decision Letter 0]

11 Nov 2021

Response to Reviewers 

We appreciate all the editor and reviewers’ comments and suggestions provided for our article entitled “COVID-19 Vaccine Hesitancy among Ethiopian Healthcare Workers. Manuscript ID: PONE-D-21-31552”. We hope to have addressed all concerns appropriately in our point-by-point response. All modifications are shown in the revised manuscript attached with file name of “Manuscript” for manuscript without track change and “Revised Manuscript with Track Changes”. Thank you for the valuable comments and we hope the editor and reviewers will be satisfied with our responses. 

Sincerely,

Atalay Mulu Fentie, on behalf of all authors

S.No Query by Response 

1. Editor 

1. 1.1. Please ensure that your manuscript meets PLOS ONE's style requirements, including those for file naming. Thank you very much. We updated the manuscript and made sure it met all of the standards. We hereby affirm that the manuscript is prepared as per the PLOS ONE requirements. 

2. 1.2. PLOS ONE does not copy edit accepted manuscripts. To that effect, please ensure that your submission is free of typos and grammatical errors. All authors did an independent review of the manuscript for any grammatical or spelling errors and made necessary modifications.

3. 1.3. We note that you have indicated in your manuscript that data from this study are available upon request. PLOS only allows data to be available upon request if there are legal or ethical restrictions on sharing data publicly. Thank you. We have updated the data availability statement as per your comments. Our data set is de-identified and the minimal anonymized data set has been uploaded as a supporting information file with the file name "S2 File." 

4. 1.4. Please include captions for your Supporting Information files at the end of your manuscript, and update any in-text citations to match accordingly. We have included the supporting information files as per the PLOS ONE guideline. 

2. 2. Reviewer 1- Dr. Qasim Mehmood 

General Comments: The manuscript is well-structured overall. Minor grammatical errors can be seen throughout. It is suggested to run the manuscript over some grammar checking app or website like Grammarly. Thank you so much for your comments, and grammatical and spelling error corrections you made on our manuscript. We did both the grammarly application to correct the errors and also all the authors independently reviewed the manuscript and made all the necessary corrections. 

Abstract: In the line 36, change the word 'nurse' to 'nurses' Thank you for the comment and it is well taken. 

Methods: In the line 109, change the word 'facilitates' to 'facility' Thank you for the comment and it is well taken.

Results: 

In the line 144, change 'medical condition' to 'medical conditions' Thank you for the comment and it is well taken. 

In the line 182, change 'printed medias' to 'printed media' Thank you for the comment and it is well taken. 

In the line 189, change the phrase 'Only few' to 'Only a few' Thank you for the comment and it is well taken. 

In the line 229, change the phrase 'trust in science' to 'trust science' Thank you for the comment and it is well taken. 

Discussion: 

Spelling mistakes and grammatical errors need correction Thank you for the comment and it is well taken. 

In the line 261, change 'different informations' to 'different pieces of information' Thank you for the comment and it is well taken. 

In the line 268, rephrase the sentence 'A relatively higher concerns' Thank you for the comment and it is well taken. 

In the line 272, correct the spelling of 'speed' Thank you for the comment and it is well taken. 

In the line 275, change 'inadequately communication' to 'inadequate communication' Thank you for the comment and it is well taken. 

In the line 277 and line 300, correct the spelling of 'hesitancy' Thank you for the comment and it is well taken. 

In the line 284, correct the spelling of 'empathetic' Thank you for the comment and it is well taken. 

In the line 299, change 'Incontrary' to 'In contrary' Thank you for the comment and it is well taken. 

In the line 303, change the word 'mirrors' to 'mirror' Thank you for the comment and it is well taken. 

In the line 311, change the word 'limitation' to 'limitations' Thank you for the comment and it is well taken. 

Conclusion: Well-written overall but few grammatical mistakes need correction Thank you for the comment and it is well taken. 

In the line 321, change the word 'belief' to 'believe' and change the phrase 'superior over vaccination' to 'superior to vaccination' Thank you for the comment and it is well taken. 

3. Reviewer 2: 

The author states: “Healthcare providers who were present at the time of data collection and willing to participate were included in the study.” Was data collected once or more than one in the same location? On particular days/times, etc? Further information is needed. Thank you so much. Since we allocate predetermined sample size to all health facilities, data were collected on a single or multiple dates depending on the number of HCWs working in the respective healthcare facility and data collection continued until the required sample size is reached. In the manuscript under “Sample size and participant recruitment” we included the following statement “We purposively selected a total of 18 hospitals (10 public and eight private hospitals) and predetermined sample size was allocated to all health facilities. Participants who met the study's inclusion criteria were invited to participate, and data collection continued until each health facility achieved the requisite sample size. 

 Indicate what table the results are in before describing them (e.g., Table 6 results). Thank you so much and your comment is well taken. We indicate tables before the narration statement.

Were vaccines not yet available to Ethiopia from March to July 2021? It would be useful to describe a little bit more about the context, since the questions were asking about whether one planned to receive a vaccine versus whether they had received a vaccine. During data collection COVID-19 vaccination was not started in Ethiopia. Now we have included the same on Study design and participants section of the method as follows “An institutional-based cross-sectional survey was conducted from March 01 to 10, 2021. Ethiopia got the first shipment of the vaccine on March 6 and vaccination of HCWs was launched on March 13, 2021, which was after our data collection. We have also include more clarification about vaccine first shipment and launch in the introduction section of the manuscript. The initial study period March to July 2021, which was stated on the first submission was meant to include data collection period as well as statistical analysis and write-up. However, now as per your suggestion we amended the study period to consider the data collection time only.

---

## [Editor Report · Decision Letter 1]

29 Nov 2021

COVID-19 vaccine hesitancy among Ethiopian healthcare workers

PONE-D-21-31552R1

Dear Dr. Fentie,

We’re pleased to inform you that your manuscript has been judged scientifically suitable for publication and will be formally accepted for publication once it meets all outstanding technical requirements.

Kind regards,

Stephan Doering, M.D.

Academic Editor

PLOS ONE

---

## [Editor Report · Acceptance letter]

9 Dec 2021

PONE-D-21-31552R1 

COVID-19 vaccine hesitancy among Ethiopian healthcare workers 

Dear Dr. Fentie:

I'm pleased to inform you that your manuscript has been deemed suitable for publication in PLOS ONE. Congratulations! Your manuscript is now with our production department. 

Kind regards, 

on behalf of

Professor Stephan Doering 

Academic Editor

PLOS ONE